# Footprints of COVID-19 on Pollution in Southern Spain

Eszter Wirth [1,*], Manuel Alejandro Betancourt-Odio [2], Macarena Cabeza-García [3] and Ana Zapatero-González [2]

1 Department of Economics, Universidad Pontificia Comillas, 28015 Madrid, Spain
2 Department of Quantitative Methods, Universidad Pontificia Comillas, 28015 Madrid, Spain
3 Investment Banking, Deutsche Bank, 28015 Madrid, Spain
* Correspondence: ewirth@comillas.edu

**Abstract:** Background: Many annual deaths in Spain could be avoided if pollution levels were reduced. Every year, several municipalities in the Community of Andalusia, located in southern Spain, exceed the acceptable levels of atmospheric pollution. In this sense, the evolution of primary air pollutants during the March–June 2020 lockdown can be taken as reliable evidence to analyze the effectiveness of potential air quality regulations. Data and Method: Using a multivariate linear regression model, this paper assesses the levels of $NO_2$, $O_3$, and $PM_{10}$ in Andalusia within the 2017–2020 period, relating these representative indices of air quality with lockdown stages during the pandemic and considering control variables such as climatology, weekends, or the intrusion of Saharan dust. To reveal patterns at a local level between geographic zones, a spatial analysis was performed. Results: The results show that the COVID-19 lockdown had a heterogeneous effect on the analyzed pollutants within Andalusia's geographical regions. In general terms, $NO_2$ and $PM_{10}$ concentrations decreased in the main metropolitan areas and the industrial districts of Huelva and the Strait of Gibraltar. At the same time, $O_3$ levels rose in high-temperature regions of Cordoba and Malaga.

**Keywords:** $NO_2$; $PM_{10}$; $O_3$; COVID-19; pollution; lockdown

## 1. Introduction

The global spread of COVID-19 became one of the current century's most significant epidemiological and economic emergencies, with worldwide infection cases of over 617 million and deaths exceeding 6.5 million as of September 2022 [1]. At first, the virus spread swiftly in Spain owing to the lack of awareness by authorities and citizens. Many neglected recommended social distancing measures after detecting the earliest COVID-19 outbreaks in late February. This resulted in overwhelmed intensive care units in conjunction with a quick escalation of deaths and led to a nationwide lockdown from 14 March 2020 until 20 June, deemed one of the strictest confinement systems in Europe [2]. The lockdown, encouraged by scientific experts [3], was successful at restraining new infections and at flattening the epidemic's curve [4], but resulted in the cancellation of mass events, closure of educational institutions, industrial factories, bars, restaurants, and shops (except those selling essential goods, such as supermarkets and pharmacies). Spain's GDP fell by 10.8% in 2020, one of the most affected economies by the pandemic due to its heavy reliance on services [5].

This strict quarantine regime and mobility restrictions offer an unprecedented opportunity to grasp the impact of anthropogenic air pollution reduction and simulate the possible outcomes of policies destined to cut traffic emissions and other polluting activities. Thus, many research papers have been published on a global scale to monitor the impact of COVID-19 induced lockdowns on air quality, mainly focusing on large urban settlements: South Asia [6–9], the Middle East [10], North America [11,12], South America [13–15], Western Europe [16,17], and Central–Eastern Europe [18–20]. Many recorded significant drops in NO, $NO_2$, $SO_2$, and $PM_{10}$ pollutants, but they found increases in tropospheric $O_3$ even

after adjusting for meteorological factors. This latter phenomenon could be attributable to the "lockdown effect", similar to an intense "weekend effect" and complex VOCs/NOx–$O_3$ chemistry in urban and rural areas. Moreover, PM and NOx levels recorded significant improvements in developing countries such as China and India [6]. At the same time, North America and Western Europe saw more modest ameliorations in air quality, mainly limited to NOx, but not PM [21–23]. This may be explained by particulate matter's long distance transport by winds within Europe and the lack of disruption in sectors such as agriculture, power generation, and residential energy use during the lockdown. Moreover, PM concentrations stayed fairly constant in Central–Eastern Europe [24] or even increased in some major Polish cities due to the combustion of fuels in furnaces for heating—mostly coal [18,20].

Research articles also focused on major Spanish cities, such as Madrid and Barcelona [25,26], Valencia [27], A Coruña and Vigo [28], and a multi-city analysis [29–31]. Authors documented intense reductions in NO, $NO_2$, and $SO_2$; milder decreases in CO, $PM_{10}$, and $PM_{2.5}$ coupled with alarming increases in $O_3$ concentrations in some areas. Studies at rural sites were fewer [32,33] and suggest a less noticeable improvement in air pollution at peripheral sites compared to urban settlements.

To our knowledge, only Hidalgo-García and Arco-Díaz [34] focused on lockdowns in the Community of Andalusia (CA or Andalusia from now on), selecting the eight provincial capitals of the region and using both satellite and ground-based air quality data. They found significant reductions in $PM_{10}$, $NO_2$, and CO concentrations in inland cities during the day and at night, while $SO_2$ mainly dropped in coastal cities, which can be attributed to the stoppage in maritime transport and the effect of sea breezes. As for tropospheric $O_3$, levels increased in inland and coastal urban areas, especially in coastal cities during the daytime. Moreover, the authors found drops in land surface temperature and surface heat islands in both types of cities.

This paper contributes to the literature along several dimensions. First, when analyzing the effects of the lockdown on air pollution, much of the focus has been placed on regions and countries. By collecting detailed local data from 80 air monitoring stations in a vast region of Spain, this paper documents how and to what extent the results can be heterogeneous even across adjacent sites.

Secondly, our article evaluates the impact of COVID-19 lockdowns on $NO_2$, $O_3$, and $PM_{10}$ pollutants at urban, suburban, and rural stations in the Spanish Autonomous Community of Andalusia (Southern Spain), characterized by increasingly unfavorable meteorological conditions during spring and summer: atmospheric stability with high solar radiation and temperatures, and frequent intrusion of dust from Northern Africa. The hourly data were collected from daily measurements from 80 air quality stations, 27 located in eastern Andalusia and 53 in its western part, encompassing the period between 2017 and 2020. We employed multiple regression techniques to model air pollution based on a variety of explanatory factors: meteorological variables, biomass combustion by wildfires, Saharan dust episodes, weekend effect, and, above all, the several phases of the nationwide lockdown period enforced by the Spanish government during 15 March to 20 June 2020. We claim that the lockdown's footprint was diverse across regions, no matter how adjacent they might be. This pattern suggests that the behavior of $NO_2$, $PM_{10}$, and $O_3$ pollutants within the CA is very complex.

This article is organized into five sections. Section 2 outlines the dataset alongside our methodology and data visualization techniques. Section 3 contains our main results, and Sections 4 and 5 present the discussion of the paper's main findings and contributions.

## 2. Methodology

### 2.1. Air Quality Standards in Spain: The Impact of Pollution in the Last Year

Air quality standards in Spain are regulated by EU Directive 2008/50/CE (approved by the European Parliament and the European Council on ambient air quality) and Royal Decree 102/2011 (approved by the Spanish government). However, the WHO updated its rec-

ommended air quality guidelines in 2021 [35] based on more than 500 scientific studies that showcase air pollution's multiple adverse consequences on health and biodiversity [36,37].

All polluting gases and particles experienced a substantial reduction during 2021 within Spain's territory when compared to the previous five-year period (2015–2019). Despite such improvements, air pollution levels did not manage to fall below the most recent limits set by the WHO, except for $SO_2$ [38]. Every Spanish inhabitant breathed polluted air in 2021, which means an increase of 3.1 million people compared to 2019 if the latest WHO guidelines are considered [38].

As for ground surface subject to high atmospheric pollution levels, 122,200 km$^2$ of the Spanish territory, 24% of the total, suffered vegetation and ecosystem damage based on critical values set by the current legislation (Directive 2008/50/CE and Royal Decree 102/2011) [38]. In other words, nearly a fourth of Spain's territory endured air pollution that failed to fulfill current legal limits set to protect agricultural crops and natural ecosystems. Springtime atmospheric stability was present throughout 2021, activating particulate matter pollution episodes, most of them loaded from North Africa. $PM_{10}$, $PM_{2.5}$, and tropospheric $O_3$ are the pollutants that had a major impact on Spanish citizens' health and well-being.

First, 39.7 million people (83.8% of the total population) were affected by $PM_{10}$ exceedances in 2021, based on the WHO's latest recommended annual limits (15 μg/m$^3$ in 24 h). This is an increase of 17.1 million people compared to 2019. Medical research associates short-term exposure to $PM_{10}$ with a higher risk of premature death [39]. The most affected autonomous communities by $PM_{10}$ exceedances were Andalusia, Aragon; the industrial and urban districts of Asturias, Galicia, the Basque Country, the Balearic and Canary Islands, Castile-La Mancha, El Bierzo; the urban agglomerations of Valladolid, Catalonia, the coast of Valencia and Murcia, Extremadura, southern Navarra and Ceuta and Melilla autonomous cities [38]. In addition, values surpassed the aforementioned daily limits at the air quality monitoring stations of Rinconcillo (Algeciras), Granada Norte (Granada's metropolitan area), Marbella Arco (near Malaga), and the ports of Almeria, Carboneras, and Motril, all located in Andalusia.

Secondly, high $PM_{2.5}$ pollution levels affected 44.9 million inhabitants in 2021 (94.7% of the total population), 16.3 million more than in 2019 according to the latest WHO recommended annual limit (15 μg/m$^3$ in 24 h). The only areas free from $PM_{2.5}$ exceedances were Maó-Es Castell (Menorca Island); Las Campiñas and the Ranges of Guadalajara and Cuenca; el Bierzo and the mountains at the north of Castile and Leon; Caceres city and the intermediate urban sites in Extremadura and the Basque Country's shore.

Thirdly, $NO_2$ pollution exceedances impacted the urban areas of A Coruña, Albacete, Alicante, Aviles, Barcelona, Bilbao, Cartagena, Ceuta, Córdoba, Cuenca, Elche, Gijon, Granada, Guadalajara, Huelva, Las Palmas de Gran Canaria, Leon, Logroño, Madrid, Malaga, Murcia, Ourense, Oviedo, Palma, Pamplona, Pontevedra, Salamanca, Santander, Santiago, San Sebastian, Santa Cruz de Tenerife, Sevilla, Tarragona, Valencia, Valladolid, Vigo, Vitoria, and Zaragoza. Therefore 62.2% of the Spanish population (29.5 million people) were suffering from high $NO_2$ pollution according to the latest annual WHO guidelines (10 μg/m$^3$ in 24 h), with irreversible short-run health effects caused by this gas [40].

Finally, tropospheric $O_3$ is a secondary atmospheric pollutant that tended to register high concentrations at the rural sites of A Coruña, Albacete, Alicante, Aviles, Barcelona, Bilbao, Cartagena, Ceuta, Cordoba, Cuenca, Elche, Gijón, Granada, Guadalajara, Huelva, Las Palmas de Gran Canaria, Leon, Logroño, Madrid, Malaga, Murcia, Ourense, Oviedo, Palma, Pamplona, Pontevedra, Salamanca, Santander, Santiago, San Sebastian, Santa Cruz de Tenerife Sevilla, Tarragona, Valencia, Valladolid, Vigo, Vitoria, and Zaragoza. It is estimated that 96.1% of the total Spanish population was subject to $O_3$ exceedances in 2021 based on the latest WHO guidelines (60 μg/m$^3$ during peak season, the average of daily maximum 8 h mean ozone concentrations), 5.3 million more than in 2019 [34]. $O_3$ is known for its severe damage to the respiratory system and the increase in premature death risk [41].

High atmospheric pollution episodes within the Spanish borders in the past five years entice to reflect upon the current legislation that regulates allowable contamination limits relative to the previously mentioned pollutants. More restrictive limits in accordance with the new WHO guidelines would contribute to more effective environmental policies.

*2.2. Study Area*

Andalusia is one of the seventeen Spanish autonomous communities, located in the southernmost part of the Iberian Peninsula. Its 87,268 km$^2$ occupies 17.3% of Spain and is divided into eight provinces: Almeria, Cadiz, Cordoba, Granada, Jaen, Huelva, Malaga, and Sevilla. It is bordered by the Spanish communities of Extremadura, Castilla-La Mancha to the north, Murcia to the east, and Portugal to the west. To the southwest, the Atlantic Ocean washes the coasts of the Huelva and Cadiz provinces. To the southeast, it is the Mediterranean Sea that meets the shores of Cadiz, Malaga, Granada, and Almeria provinces.

Andalusia is the most highly populated community: at the end of 2021, it was inhabited by 8,472,407 people, with most living in the provinces of Sevilla (1.95 million), Malaga (1.70 million), and Cadiz (1.24 million) [42]. The inhabitants are unevenly distributed across the community, as more than 80% reside in urban areas. Andalusia's economy is the third largest in Spain, representing 13.4% of the national GDP. Despite being one of the poorest regions in terms of GDP per capita, its income per capita grew above the Spanish rate between 1995 and 2021. During 2020 the community's total GDP fell by 10%, which was 0.8 percentage points milder than the plunge of the national GDP [43].

The average altitude is 503 m above sea level (m.a.s.l.), but it is highly heterogeneous: 50% of the territory is mountainous and presents 46 peaks exceeding 1000 m.a.s.l. On the other hand, at the center of Andalusia is the 350 km long Guadalquivir Valley, which is only 100 m.a.s.l. and is mainly used for agricultural activities. This significant depression separates the Iberian Massif from the Baetic Ranges. These latter consist of three mountainous ranges aligned in a southwest–northeast direction: (i) the Penibaetic System in the southeast, parallel and close to the coast from Malaga to Murcia, made up of the highest peaks; (ii) the Subbaetic System, which occupies a central position within the Baetic Ranges; and (iii) the Prebaetic System, the northernmost feature of the ranges, which extends from the eastern points such as Jaen and Granada towards the North of Alicante. Two territories can be identified to the south of the Guadalquivir Valley: the Mediterranean coastal strip and an interior strip between the Subbaetic and the Penibaetic systems. The latter is a transition zone influenced both by air circulation from the Guadalquivir Valley and Mediterranean Sea breezes that can penetrate from the south through the paths between Malaga–Antequera, from Motril towards Granada, or through Sierra Nevada–Sierra Baza, from Almeria towards Guadix. In the coastal strip sea breezes dominate, favored by south-oriented mountainside heating caused by solar radiation. Air pollution dispersion could be boosted more or less by mountainous barrier proximity, which lengthens or compresses the range of sea breezes. Winds that go down parallel to the coast towards the Gibraltar Strait are frequent in the eastern end of the strip and could scatter O$_3$ pollution [44].

Owing to its complex topography of high mountains, inland depression, and a 950 km coastline, the region presents several background subclimates. The Mediterranean climate dominates in the community, except for Granada and Almeria, which share a semiarid climate: Almeria is characterized by several desert-like zones, and Granada by cold areas at mountainous sites [45].

The region is part of the Western Mediterranean Basin (WMB), one of the most exposed areas to PM and O$_3$ pollution in Europe [46] due to its Mediterranean-type climate featuring high-temperature summers, low-speed winds, and high solar radiation. Moreover, Saharan dust intrusions from Northern Africa are frequent phenomena in summer and during winter–spring, contributing to significant increases in PM levels [47,48]. The Iberian Peninsula is also subject to sea breeze dynamics that transport O$_3$ towards inland regions and are responsible for PM emissions near the shore [49].

Andalusia is particularly exposed to unfavorable pollution dynamics due to its closeness to the African Continent, mild winters, high-temperature summers, scarce precipitation, and intense solar radiation compared to the rest of the Iberian Peninsula. $PM_{10}$ concentrations are above the Spanish and mainland European averages due to high traffic and industrial emissions, the high frequency of Saharan dust outbreaks, precipitation scarcity, and dry soils that favor the resuspension of particulate matter [50–53].

Moreover, high atmospheric pressures, summer heat, solar radiation, and sea-valley breeze circulations also favor high $O_3$ episodes in Andalusia, particularly in the Guadalquivir Valley, southwestern (SW) Spain [54]. This valley acts as a natural channel enabling the circulation of polluted air masses from the Atlantic shore towards the inner regions of the community [55]. Prominent is the transport of polluted air from the bay of Algeciras and the petrochemical district of Huelva city towards Sevilla, Cordoba, and Jaen, where these pollutants mix with emissions from road traffic, local industries, and agricultural activities [56,57]. In spring, winds mainly blow from the northwest and southwest along the valley's axis. Atlantic air mass loads are also frequently channeled towards the valley from the southwest. During summer, most winds originate from the southwest [54]. Furthermore, synoptic conditions, such as sea–land breezes, high surface pressure in the north and west of the Iberian Peninsula, and anticyclonic conditions, favor the accumulation of pollutants near the surface, preventing their mixture with the air above [58,59]. According to Millán-Martínez [60] and Diéguez et al., [44], aged air masses from the WMB may enter the Guadalquivir Valley through the Gibraltar Strait and increase pollution levels. Additionally, Pay et al., [61] argue that the Guadalquivir Valley is among the most affected Spanish regions by $O_3$ stagnation conditions, in conjunction with the Mediterranean coast and the north–northeast of the Iberian Peninsula.

High and persistent $O_3$ levels are also present in Andalusia's southeastern (SE) region, as in Almeria province, despite scarcely populated areas and lower $NO_x$ road traffic emissions. López-Muñoz et al. [62] attribute this situation to the so-called "mirror effect". Part of $O_3$-rich air masses originate from more populated coastal eastern agglomerations and the WMB, and afterward are carried by eastern winds towards Almeria. Moreover, western winds also transport polluted air from the Strait of Gibraltar and the Atlantic Ocean.

Even though the industrial sector represents a low share of the regional GDP, two major industrial hotspots lie in the SW, in Ria of Huelva, at the confluence of the Odiel and Tinto rivers that flow into the Atlantic Ocean, and the Strait of Gibraltar, featuring metallurgic, petrochemical, and fertilizer production, which contribute to high PM emissions. The Bay of Algeciras, located next to the Strait of Gibraltar, also shows high industrial activity emissions, exacerbated by freight transport passing through the strait and high traffic [63,64]. The wind directions in this area are affected by land–sea breeze circulations and the topography of the Tinto and Odiel rivers. Air masses emitted at Huelva are often transported across the Tinto River valley towards the northeast, including remote zones next to areas of high ecological value such as Doñana National Park and the Odiel Marshes [50].

Bailén lies in northeastern Andalusia, within the Guadalquivir Valley, and is frequently affected by low air quality due to its ceramic factories (brick and pottery), which are sources of $PM_{10}$ and $SO_2$ pollution [65]. Coke is the major energy source and mixes with other low-grade fuels. On the southeastern coast, the Carboneras industrial district stands out in Almeria province, made up of a coal-based thermal plant, oil refinery, cement factory, desalination plant, and two industrial seaports. This industrial area contributes to severely high $O_3$, $NO_x$, PM, and $CO_2$ emissions on CA's southeastern coast.

Public transport is deficient in the CA, so inhabitants must commute long distances using private vehicles. This results in dense road traffic and high road emissions within the CA [66]. Sevilla and Malaga are among the provinces with the highest number of road vehicles, after Madrid, Barcelona, Valencia, and Alicante, which favor NO, $NO_2$, and $PM_{10}$ emissions [67]. Moreover, Sevilla city has several emission hotspots as its harbor and the PERSAN chemistry industry [68]. Airport traffic is present in Malaga, Cadiz,

Almeria, Sevilla, and Cordoba. According to the Spanish airport operator Aena [69], Malaga airport received 7.4% of passengers in Spain during 2020, registering a quota just below those of Madrid, Barcelona, and Palma de Mallorca. Malaga is the major coastal city in Andalusia and an important tourist destination together with Marbella city, located within the same province. Both increase their population during summer and are surrounded by mountainous formations to the north and are affected by southeast and northwest winds due to sea-land and land-sea breezes.

The city of Cordoba has a nearby smelter district, and Granada is a peculiar city surrounded by mountains up to 3500 m.a.s.l. of Sierra Nevada that trap pollutants inside its natural basin [70]: its topography favors temperature inversion and low wind speed that enable the accumulation of pollutants and their permanence near the surface. Furthermore, Granada's total car fleet is high (775 cars per 1000 inhabitants), so road traffic is the primary source of pollution in the absence of nearby industrial districts [71,72].

Previous studies have shown the relationship between industrial activities, traffic and the level of $PM_{10}$. In Andalusia, high pollution levels have been reported in industrial cities, including $PM_{10}$ between the pollutants with exceeding levels. The different sources of $PM_{10}$ in Andalusia are soil dust, road traffic, industrial emissions, oil combustion, sea salt, secondary nitrate, and secondary sulfate [73]. Soil dust includes road dust, construction dust, and African dust, whereas oil combustion emissions are due to industrial combustion, coke combustion, and sea freight emissions from ships. The transportation and energy industry explains the presence of secondary sulfate; which is formed in the atmosphere from other sulfur-containing compounds under mechanisms that involve photochemical processes, including the operation of several industrial processes such as at refinery plants and coal-fired and thermal power plants.

The sources of $PM_{10}$ weighed differently in the function of the geographical situation of the station where it is measured [69]. It has been reported that in urban industrial areas of Seville, the largest sources of $PM_{10}$ are soil dust (26%) and traffic (25%). In contrast, the metropolitan industrial area of Algeciras showed the highest concentration of secondary nitrate, which can be explained by the presence of a large oil refinery, fuel, and coal-fired plants, and Algeciras' seaport. The suburban station analyzed in Huelva showed $PM_{10}$ from road traffic (18%), industry (17%), and secondary sulfate (15%), where the copper smelting industry and mining could explain the presence of secondary sulfate. The urban station in Jaen and the traffic station in Almeria showed traffic as the main source of $PM_{10}$. In the rural area of Malaga, the sources of $PM_{10}$ are soil dust (35%), road traffic emission (26%), oil combustion (16%), and secondary sulfate (15%).

Concerning $NO_2$ and $O_3$, there is no evidence of research that have described in quantitative detail their individual sources in Andalusia, perhaps because of the complex process of their formations. The high temperatures and high solar radiation with the orography of Andalusia and the effect of anthropogenic activities of several Andalusian highly populated cities and industrial complexes are the main reasons for the levels of tropospheric ozone that characterize Andalusia. If we consider the daily cycle of the concentrations of these pollutants, in urban and suburban areas, the highest ozone levels are reached in the early afternoon. At the same time, the maximum peaks of $NO_2$ and NO are reached during the early hours of the morning and the last hours of the evening. The behavior of these pollutants in rural areas depends on natural sources, and they present no hourly peak. The weekend effect on these pollutants is only present in areas with high traffic emissions [58].

### 2.3. Data

To monitor and control air pollution in the CA, hourly data from 80 air quality monitoring stations were used, 27 located in eastern Andalusia (Almeria, Granada, Jaen, and Malaga) and 53 in its western part (Cadiz, Cordoba, Huelva, and Sevilla).

The air monitoring stations are designed to produce a continuous flow of indicators regarding air pollution within the CA. This information is then used for surveillance,

forecasting, and information purposes. The surveillance system consists of automatic remote stations, which are spread all over the territory of the CA, as displayed in Figure 1a,b (the latter represents an enlarged version of the area outlined by the blue line in Figure 1a). The air quality monitoring stations are of four different kinds: urban (U), situated in urban communities; industrial (I), situated in the industrial zone; remote (R), situated in rural areas; and urban traffic (UT), situated in metropolitan areas. Each air monitoring station is classified according to the area's features (traffic levels, population, distance to the capital's city center, etc.) and under no circumstances is biased among data collected at different stations according to this classification.

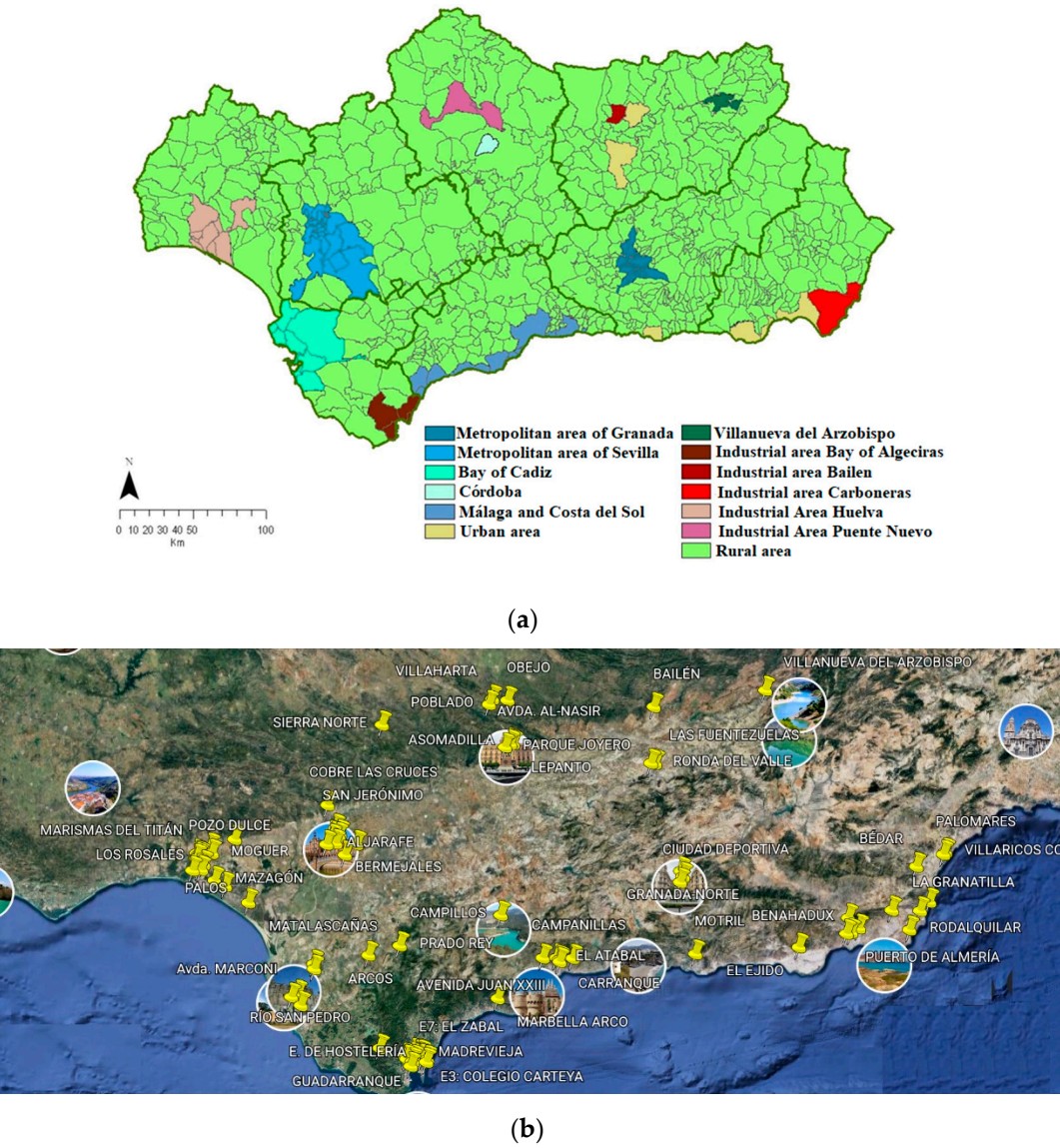

**Figure 1.** (**a**) Andalusia zoning map and (**b**) distribution of air quality monitoring stations.

The Ministry for the Ecological Transition of Spain issues the data [74]. The sample period extends from 1 January 2017 to 31 December 2020, i.e., t = 1460 observations for each air station–pollutant analyzed.

The daily average concentrations for $NO_2$, $PM_{10}$, and $O_3$ were calculated throughout the 2017–2020 period using the hourly concentration of the different air pollutants.

The measurement techniques of these three pollutants analyzed and reported in this article are defined in section c of Annex V of Royal Decree 102/2011. For $NO_2$, the reference method is defined in norm UNE-EN 14211:2013 "Environmental Quality.

Normalized measurement method of $NO_2$ and $NO_x$ concentration by chemiluminescence". This method generates NO and $O_3$ reaction in a chamber to create excited $NO_2$. When $NO_2$ molecules return to their ground state, they emit radiation which is proportional to the initial concentration of NO. The measurement and processing of data are identical across stations.

The method used for $O_3$ is described in norm UNE-EN 14625:2013 "Air environment. Normalized measurement method of ozone concentration by ultraviolet photometry" [75]. In this case the ultraviolet absorption is used: $O_3$ absorbs part of the UV radiation when the sampled air flow is passed through a normalized air detector. $O_3$ concentration can be estimated according to its attenuation.

In order to provide valid usable data both for $NO_2$ and $O_3$, all fixed air quality stations should guarantee during the studied period of time at least 90% of the possible data capture in summer and 75% in winter, and the uncertainty level should not surpass 15% [75].

The measurement method for $PM_{10}$ and $PM_{2.5}$ is described by Decree UNE-EN 12341:2015 "Environmental Air. Normalized gravimetric measurement for $PM_{10}$ and $PM_{2.5}$ mass concentration in suspension". Gravimetric analysis of PM is based on the collection of particulate matter through filters at a known flow rate for a specific duration. Any PM present is retained on the filter. A cutting head later selects the reference particulate matter's fraction ($PM_{10}$ or $PM_{2.5}$). The concentration is obtained as the difference between the filter weight after and before sampling, divided by the total volume. In order to provide valid usable data, all fixed air monitoring stations should guarantee at least 90% data capture through the time–space and the uncertainty level should not surpass 25% [75].

In addition, meteorological data were downloaded from the Open Data platform of the Spanish Meteorological Agency (AEMET) via its Application Programming Interface (API). To associate meteorological stations with air quality monitoring stations, the closest one (using great circle distance) was chosen to obtain the meteorological variables of interest where air quality monitoring stations are located: temperature (°C, minimum, mean, maximum), precipitation (mm), wind speed (kph), Saharan dust episodes, and biomass combustion. Dust episodes and biomass combustion are recorded at the regional level by the Ministry for Ecological Transition and Demographic Challenge of Spain and are publicly available [74].

*2.4. Econometric Approach*

We model the concentration of pollutant $k$ at station $s$ as a function of explanatory variables:

$$\ln(P_{skt}) = \alpha_{sk} + \beta_{sk}LD_t + \gamma_{sk}X_{skt} + \varepsilon_{skt} \tag{1}$$

where $P_{skt}$ is the average daily level of pollutant $k$ at station $s$ on day $t$; $LD_t$ is a vector of binary variables that takes value one if the corresponding lockdown phase is in place, zero otherwise; X is an entire vector of control variables; and $\varepsilon_{skt}$ is an error term. Parameter vector $\beta_{sk}$ (marginal coefficients ) captures the effect of each *LD* phase on pollutant $k$ at station s and is the object of detailed analysis. The intercept and the effect of the remaining independent variables are captured by $\alpha_{sk}$ and $\gamma_{sk,\,t}$, respectively.

We take logarithms of pollution levels because their distribution is highly right-skewed. The second reason is that logs can interpret all the estimated effects in percentage terms. A separate equation is estimated for the pollutants considered ($NO_2$, $PM_{10}$, and $O_3$).

Weather patterns (temperature, precipitation, wind speed) can considerably affect ground-level pollutant concentrations, compromising the LD's observable effects [76,77]. Therefore, meteorological factors are considered in the regression model. Specifically, vector X includes: the daily maximum temperature (in °C), the daily minimum temperature (in °C), the daily average wind speed (in kph), and the daily rainfall (in mm).

Like many other regions, Spain can be affected by Saharan dust intrusion episodes and biomass combustion, influencing pollution levels. To capture this potential effect, X also includes a set of dummy variables that takes value 1 in the case of a Saharan dust

episode or biomass combustion. Vector X also includes a binary variable for weekend days (Saturdays or Sundays), as economic activity and mobility are less intense.

Finally, Equation (1) is fitted to the data through ordinary least squares (OLS), a procedure that eliminates from the effects of each LD phase ($\beta_{sk}$), the effects of the remaining independent variables incorporated in X. In other words, the LD phase effects reported in the paper are ceteris paribus, i.e., keeping all those other factors constant.

### 2.5. Lockdown Period Determination

Andalusia's first COVID-19 outbreak was detected on 26 February 2020. Faced with overburdened intensive care units and the swift increases in COVID-19 related deaths, the Spanish government declared a nationwide lockdown through the "state of alarm" on 13 March, which was among the strictest confinement systems in Europe and lasted until 20 June. Thus, we define the variable "Complete lockdown" from 15 March to 20 June.

On 31 March, the lockdown was toughened further to reduce the virus' diffusion, and all nonessential activities were prohibited until 13 April. The toughened lockdown was lifted on 13 April, and industrial activities could resume. On 4 May, children and adults were allowed to leave their homes and carry out outdoor walks and jogging. After 5 May, a de-escalation schedule was proposed to lift the lockdown in four phases gradually, according to each Spanish community's epidemiological condition. On 25 May, small shops selling nonessential goods could open, and the restaurants' terraces were operative with 30% capacity and tables separated by at least 2 m. On 8 June, the inner part of shops and restaurants reopened, and specific cultural events could be hosted anew. The "state of the alarm" was finally abandoned on 20 June. On 21 June, the inhabitants of Spain could move freely between the country's regions, and each autonomous community was allowed to implement its own measures to control the virus' spread.

In our regression model, we define the following phases of the nationwide lockdown in the CA:

- Pre-lockdown: 1 January 2017–14 March 2020.
- Complete lockdown: 15 March–20 June 2020.
- Strict lockdown (phase 0): 15 March–10 May 2020.
- Lockdown (phase 1): 11–24 May 2020.
- Lockdown (phase 2): 25 May–7 June 2020.

## 3. Results

### 3.1. Meteorological Analysis

Significant differences were registered between March and June for 2017, 2018, 2019, and 2020. According to AEMET, 2017 was extremely hot in the Iberian Peninsula. Moreover, a +2 °C rise in the mean annual temperature was recorded in the central area of Andalusia. This temperature anomaly was more pronounced in May and June. The year 2018 was cold or extremely cold in Andalusia; March, and May were the months of the studied period with lower mean temperatures, ranging from 1.7 to 3.8 degrees below the average mean temperature. The mean temperature during the studied period for 2019 changed from hot in March to cold in April and June in Andalusia. Finally, 2020 was considered very hot in Spain, with corresponding high mean temperatures in Andalusia between March and June, with a peak in May.

As for precipitation, in 2017, there was a significant variation between March and June, March and April being very wet (with an increment of 241% over mean rainfall values for Huelva in March), and May and June were very dry in comparison with mean precipitation values. The year 2018 was considered wet in Andalusia for the considered months of the study, with a monthly precipitation value for March that exceeded the mean values, and an outstanding increment of 500% in Granada. In contrast to the previous years, in 2019, the weather was deemed dry or very dry during the studied four-month period, while 2020 was considered a year with a high amount of precipitation, although not as abundant as in 2017.

When the wind is considered, during the year 2017, there were episodes of strong winds in Andalusia, as well as in 2020, when the deep depressions Karine, Myriam, and Norberto affected Spain in March, and strong winds blew in the coastal area in April (139 kph in Cadiz) and May.

These changes in daily mean precipitation levels in various areas of the CA during the past three years require controlling meteorological variables to properly analyze the lockdown measures' impact on pollution levels [78].

### 3.2. Regression Model Analysis

In this section, the relation between the explanatory variables of the model and the pollutants levels is characterized. The results are reported in Tables 1 and 2. The mean values of the estimated coefficients have been averaged across stations to identify general patterns. The number of samples differs for each pollutant and area since not all the stations monitor identical specific particles. The mean standard error at the bottom of the table indicates that the sampling results represent the population. It ranges from 0.002 to 0.082 for the different pollutants in the considered areas of the study.

**Table 1.** Regression results from pollutants. Columns 3 to 5 display the estimated mean coefficients between the regression models of each independent variable (independent variable column) relative to each dependent variable (pollutant) in southeastern Andalusia.

| Independent Variables | Description | $NO_2$ | $O_3$ | $PM_{10}$ |
|---|---|---|---|---|
| **T max (°C)** | mean | −0.006 | 0.017 | 0.010 |
| | Negative and significant (%) | 53.85% | 0% | 25% |
| | Positive and significant (%) | 34.62% | 95.65% | 62.50% |
| **Wind speed (m/s)** | mean | −0.125 | 0.078 | −0.067 |
| | Negative and significant (%) | 100% | 0% | 87.50% |
| | Positive and significant (%) | 0% | 100% | 0% |
| **Rainfall (mm)** | mean | 0.006 | 0.001 | −0.009 |
| | Negative and significant (%) | 57.69% | 17.39% | 43.75% |
| | Positive and significant (%) | 7.69% | 26.09% | 0% |
| **Complete lockdown (1: yes, 0: no)** | mean | −0.327 | 0.069 | 0.003 |
| | Negative and significant (%) | 57.69% | 13.04% | 37.50% |
| | Positive and significant (%) | 3.85% | 34.78% | 31.25% |
| **Strict lockdown (phase 0) (1: yes, 0: no)** | mean | −0.324 | 0.093 | −0.337 |
| | Negative and significant (%) | 61.54% | 4.35% | 81.25% |
| | Positive and significant (%) | 11.54% | 43.48% | 0% |
| **Lockdown (phase 1) (1: yes, 0: no)** | mean | −0.072 | 0.117 | −0.350 |
| | Negative and significant (%) | 26.92% | 4.35% | 68.75% |
| | Positive and significant (%) | 7.69% | 26.09% | 0% |
| **Lockdown (phase 2) (1: yes, 0: no)** | mean | 0.136 | 0.021 | −0.185 |
| | Negative and significant (%) | 34.62% | 0% | 18.75% |
| | Positive and significant (%) | 3.85% | 17.39% | 0% |
| **Saturday (1: yes, 0: no)** | mean | −0.147 | 0.031 | −0.026 |
| | Negative and significant (%) | 84.62% | 0% | 12.50% |
| | Positive and significant (%) | 0% | 43.48% | 0% |
| **Sunday (1: yes, 0: no)** | mean | −0.290 | 0.055 | −0.106 |
| | Negative and significant (%) | 92.31% | 0% | 93.75% |
| | Positive and significant (%) | 0% | 69.57% | 0% |
| **Saharan dust (1: yes, 0: no)** | mean | −0.001 | 0.050 | 0.261 |
| | Negative and significant (%) | 38.46% | 8.70% | 0% |
| | Positive and significant (%) | 34.62% | 78.26% | 100% |
| **Biomass combustion (1: yes, 0: no)** | mean | 0.100 | −0.030 | −0.085 |
| | Negative and significant (%) | 3.85% | 34.78% | 56.25% |
| | Positive and significant (%) | 61.54% | 0% | 6.25% |
| **R squared** | mean | 0.061 | 0.039 | 0.002 |
| **No. of models** | | 26 | 23 | 16 |

**Table 2.** Regression results from pollutants. Columns 3 to 5 display the estimated mean coefficients between the regression models of each independent variable (independent variable column) relative to each dependent variable (pollutant) in southwestern Andalusia.

| Independent Variables | Description | $NO_2$ | $O_3$ | $PM_{10}$ |
|---|---|---|---|---|
| **T max (°C)** | mean | −0.007 | 0.018 | 0.008 |
| | Negative and significant (%) | 55.26% | 6.45% | 11.76% |
| | Positive and significant (%) | 23.68% | 87.10% | 70.59% |
| **Wind speed (m/s)** | mean | −0.132 | 0.073 | −0.023 |
| | Negative and significant (%) | 89.47% | 0% | 52.94% |
| | Positive and significant (%) | 2.63% | 100% | 26.47% |
| **Rainfall (mm)** | mean | 0.006 | −0.001 | −0.004 |
| | Negative and significant (%) | 13.16% | 32.26% | 44.12% |
| | Positive and significant (%) | 50% | 9.68% | 14.71% |
| **Complete lockdown (1: yes, 0: no)** | mean | −0.467 | 0.005 | −0.201 |
| | Negative and significant (%) | 78.95% | 12.90% | 58.82% |
| | Positive and significant (%) | 2.63% | 9.68% | 5.88% |
| **Strict lockdown (phase 0) (1: yes, 0: no)** | mean | −0.129 | 0.206 | −0.122 |
| | Negative and significant (%) | 42.11% | 0% | 44.12% |
| | Positive and significant (%) | 13.16% | 77.42% | 11.76% |
| **Lockdown (phase 1) (1: yes, 0: no)** | mean | 0.096 | 0.172 | −0.162 |
| | Negative and significant (%) | 7.89% | 3.23% | 44.12% |
| | Positive and significant (%) | 26.32% | 51.61% | 2.94% |
| **Lockdown (phase 2) (1: yes, 0: no)** | mean | 0.003 | 0.086 | −0.004 |
| | Negative and significant (%) | 10.53% | 0% | 8.82% |
| | Positive and significant (%) | 5.26% | 16.13% | 2.94% |
| **Saturday (1: yes, 0: no)** | mean | −0.164 | 0.035 | −0.049 |
| | Negative and significant (%) | 89.47% | 0% | 50% |
| | Positive and significant (%) | 0% | 41.94% | 0% |
| **Sunday (1: yes, 0: no)** | mean | −0.266 | 0.057 | −0.084 |
| | Negative and significant (%) | 89.47% | 0% | 73.53% |
| | Positive and significant (%) | 0% | 64.52% | 0% |
| **Saharan dust (1: yes, 0: no)** | mean | 0.080 | 0,030 | 0,266 |
| | Negative and significant (%) | 0% | 6.45% | 0% |
| | Positive and significant (%) | 55.26% | 58.06% | 100% |
| **Biomass combustion (1: yes, 0: no)** | mean | 0.074 | 0.012 | 0.033 |
| | Negative and significant (%) | 5.26% | 9.68% | 23.53% |
| | Positive and significant (%) | 57.89% | 29.03% | 50% |
| **R squared** | mean | 0.082 | 0.044 | 0.059 |
| **No. of models** | | 38 | 31 | 34 |

The effect of the complete lockdown was more pronounced on $NO_2$ (average decreases of 46.7% in the SW and 37.7% in the SE relative to the pre-lockdown period, (1 January 2017 to 14 March 2020). However, the values differ in both areas in each lockdown phase. Moreover, the share of air monitoring stations where the estimated parameter was negative and significant (*p*-value below 0.1) was higher in the SE (78.95%) compared to the SW (57.69%). The percentage of stations with positive and significant parameters in both areas was less than 4%. The increases in $O_3$ concentration during the complete lockdown were very low in both areas, ranging from 6.9% in the SE to 5% in the SW, featuring low significance rates, positive and negative, in both SE and SW areas between 9.68% and 34.78%.

The lockdown also significantly affected the concentrations of $PM_{10}$ in Andalusia but heterogeneously in the pollutant levels and their significance. While in the SW area of Andalusia, where the two major industrial districts lie, the decrease in $PM_{10}$ concentration

reached 20.1%, with a statistically significant effect in 58.82% of the stations, in the SE area, the effect on $PM_{10}$ pollution was very small and positive, merely 0.3%, with substantial heterogeneity across stations (positive and significant effect at 31.25%, negative and significant effect at 37.50%).

Furthermore, when the three phases of the lockdown are considered, significant differences were found in the pollutants' behavior. The highest reduction in $NO_2$ was registered during strict lockdown (phase 0), with decreases of 32.4% in SE Andalusia and 12.9% in the SW (significant ne effect at 61.54% and 42.11% of respective stations), and then deaccelerated during phase 1, presenting a 7.2% decrease in SE Andalusia and a 9.6% increase in the SW. However, 35% of the stations presented significant negative effect values.

The concentration of $PM_{10}$ during strict lockdown (phase 0) fell in both the SE (33.7%) and SW areas (20.1%), and significance levels were lower in the SW (44.12%) than in the SE (81.25%). Along the phases of the lockdown, the concentrations of $PM_{10}$ were slowing, presenting in the last phase a drop of 18.5% in the SE and 0.04% in the SW, with decreasing significance levels (below 20% in the SE and below 10% in the SW). The concentration of $O_3$ decreased from phase 0 to phase 2 at a slowing rate, from 9.3% in phase 0 to 2.1% in phase 2 in the SE area and from 20.6% to 8.6% in the SW. These results were more statistically significant in the SW region than the SE, although significance levels dropped along the phases and did not even surpass 20% during phase 2.

As for the remaining variables of the model, the mean daily temperature was associated with a slight drop in the level of $NO_2$: 0.6% and 0.7%, respectively, for SE and SW areas (see Tables 1 and 2), with a significant coefficient at 53.85% of the stations in the SE and 55.26% in the SW region. The daily mean temperature effect was positive upon $O_3$ and $PM_{10}$ pollution, although it was much higher and significant on the former. Hence, a 1-degree increase in daily mean temperature was associated, ceteris paribus, with an increment in tropospheric $O_3$ concentration of 1.7% in the SE and 1.8% in the SW areas. The effects were negative and significant at only 12.9% of the stations in the SW and 4.35% of the SE. The effect of $O_3$ was practically zero in the $PM_{10}$ equation, ranging from 0.8% in the SW to 1% in the SE, with greater positive significance in the SW (70.59%) than in the SE (62.50%).

Wind speed is associated with improvements in air quality when considering $NO_2$ and $PM_{10}$ concentrations. An increase of 1 m/s in wind speed induces a decrement in the pollutant levels, oscillating from 2.3% ($PM_{10}$) to 13.2% ($NO_2$) in the SW and from 6.7% ($PM_{10}$) to 12.5% ($NO_2$) in the SE with a significant negative effect (above 85% of significance) for most of the pollutants and regions. The exception to this pattern is $O_3$, given that air recirculation implies an increase in its concentration (approximately 7.5 % in both areas, with a significant result at all the stations).

The amount of rainfall had a very low impact on $NO_2$, $O_3$, and $PM_{10}$ levels. It was significant at less than 65% of the stations, being positively associated with $NO_2$ (increment of 0.6% for each mm of rainfall) and negatively with $PM_{10}$ (0.9% in SE and 0.4% in SW). However, the effect of rain on $O_3$ was very low, ranging from an increment of 0.1% in the SE area to a drop of 0.1% in the SW area (see Tables 1 and 2, respectively).

Pollution by $NO_2$ and $PM_{10}$ fell during the weekends. The effect was more pronounced for $NO_2$, particularly on Sundays, (mean reduction ranging from 15.5% on Saturdays to 27.7% on Sundays) and moderate for $PM_{10}$ (−3.75% on Saturdays to 9.5% on Sundays), with a highly significant negative effect in both areas during Sundays (73.53% in SW and 93.75% in SE). As expected, tropospheric $O_3$ levels increased by an average of 3.3% on Saturday and 5.6% on Sunday, due to the lower use of vehicles and the subsequent decrease in $NO_2$.

Finally, when natural sources of pollution are considered, Saharan dust loads worsened air quality mainly by favoring $PM_{10}$ levels, which rose 26.1% in the SE and 26.6% in the SW, both results being statistically significant at 100%. Similarly, Saharan dust also led to higher $O_3$ concentrations in the whole CA: +3% in SW and +5% in SE. However, significant

levels were noticeably higher in the SE (78.26%) when compared to the SW (only 58.06%). Nonetheless, effects on $NO_2$ were diverse, being positive in the SW (+8%) and negative in the SE ($-0.1$%), although neither result was statistically significant.

Biomass combustion had mixed impacts on $O_3$ and $PM_{10}$: in the SW; it led to mild increases in both pollutants, which contrasts with the SE, where both experienced decreases, although neither result was statistically significant. However, biomass burning worsened air quality by lifting $NO_2$ levels by 7.5% in the SW and up to 10% in the SE, registering significance levels that exceeded 60% in both cases. Between 1 January and 24 June, sixteen forest fires were documented in Andalusia, mainly in Almeria province in the SE (Cuevas de Almanzora, Terque, Níjar, Lubrin, Tabernas, Enix, and Turra), Cordoba (Guadalcázar, Fuente Obejuna, Obejo, and Montoro), Huelva (Almonaster la Real and Calañas), Granada (Padul), Malaga (Casares), and Sevilla (Aznalcollar) [79].

### 3.3. Local Analysis

In this section, we conduct a local-level analysis to establish if the differences between the level of pollutants and the critical variables of the model change across the considered regions of Andalusia. The delineated maps offer a geographic perspective in Figures 2–4, where the effect of Saharan dust is represented, detailed by the station (see Figure 2), and the effect of COVID-19 during strict lockdown (phase 0) (see Figure 3) and lockdown phase 2 (see Figure 4).

By taking the variable Saharan dust as a reference, we can contextualize the scope of pollution reduction caused by traffic restrictions during the COVID-19 lockdown when we compare it to the level of the pollutants. The panels included in Figure 2 show that the effect of Saharan dust was heterogeneous across the different provinces of Andalusia; moreover, the behavioral pattern of $NO_2$, $O_3$, and $PM_{10}$ was not coincident.

The first two panels (see Figure 2a,b) show decrements in $NO_2$ at several stations in the SW (Asomadilla, 70.71% and Avenida Al Nasir, 78.62%, both located in Cordoba province), while in the area of Gibraltar, the variation in $NO_2$ concentration was positive and relatively high, as at Palmones, 47.03% and Alcornocales, 37.49%. Both positive and negative variations in the level of $NO_2$ were recorded at the Benahadux, $-22.11$%; Motril, $+10.57$%; and Palacio de Congresos, $+15.60$% stations in the SE.

On the contrary, the presence of Saharan dust had a moderately negative effect on the behavior of $O_3$ levels, with only five stations displaying a low positive increment in the SW (1.16% in Cortijillos), which was slightly more pronounced in the SE (see Figure 2c,d). As expected, the episodes of Saharan dust had an essential impact on $PM_{10}$ concentration levels. In the SW, the effect was more pronounced in Huelva, with an increase of around 30% at most stations. The rest of the SW presented mixed behavior, with stations registering both positive (Santa Clara in Sevilla, 26.43%) and negative effects (Torneo in Sevilla, $-23.17$%; Cartuja, $-35.23$%; Algeciras, $-35.22$%). In the SE, $PM_{10}$ levels decreased at the Malaga, and Almeria stations (the episode was only registered at four air quality stations), while in Granada and Jaen, the effect of Saharan dust was very low, ranging from $-5$% to $+5$%.

We conducted a second analysis to assess the consequences of phases 0 and II of the lockdown in Andalusia when studying the variation in the concentrations of $NO_2$, $O_3$, and $PM_{10}$ in 2020.

The effect of the strict lockdown (phase 0) was highly remarkable in the registered levels of $NO_2$, (see Figure 3a) with a substantial reduction in $NO_2$ concentrations in the SW, except for the province of Cadiz (positive increments in Gibraltar). The decrease in $NO_2$ concentrations in the SE during this phase was outstanding, reaching a maximum reduction in Motril, $-88.27$% (see Figure 3b). As expected, the effect of phase 2 on $NO_2$ levels was more moderate than in phase 0, and the variations were inverse to the amount of social mobility (see Figure 4a,b).

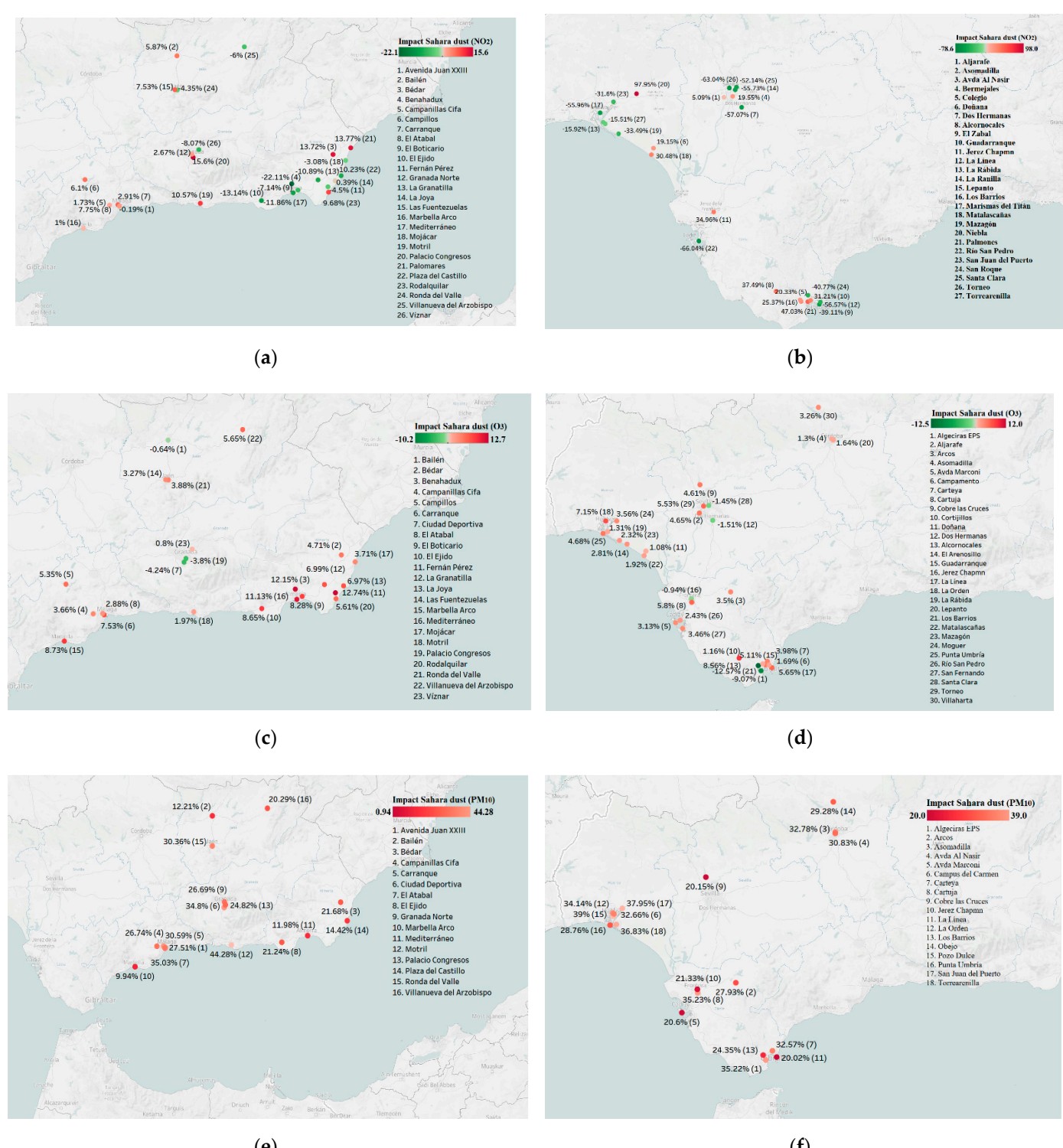

**Figure 2.** The left and right sides contain the estimates of the Saharan dust effect by air monitoring stations in the SE and SW, respectively. (**a**) Saharan dust intrusion on $NO_2$ in SE Andalusia; (**b**) Saharan dust intrusion on $NO_2$ in SW; (**c**) Saharan dust intrusion on $O_3$ in SE Andalusia; (**d**) Saharan dust intrusion on $O_3$ in SW Andalusia; (**e**) Saharan dust intrusion on $PM_{10}$ in SE Andalusia; (**f**) Saharan dust intrusion on $PM_{10}$ in SW Andalusia.

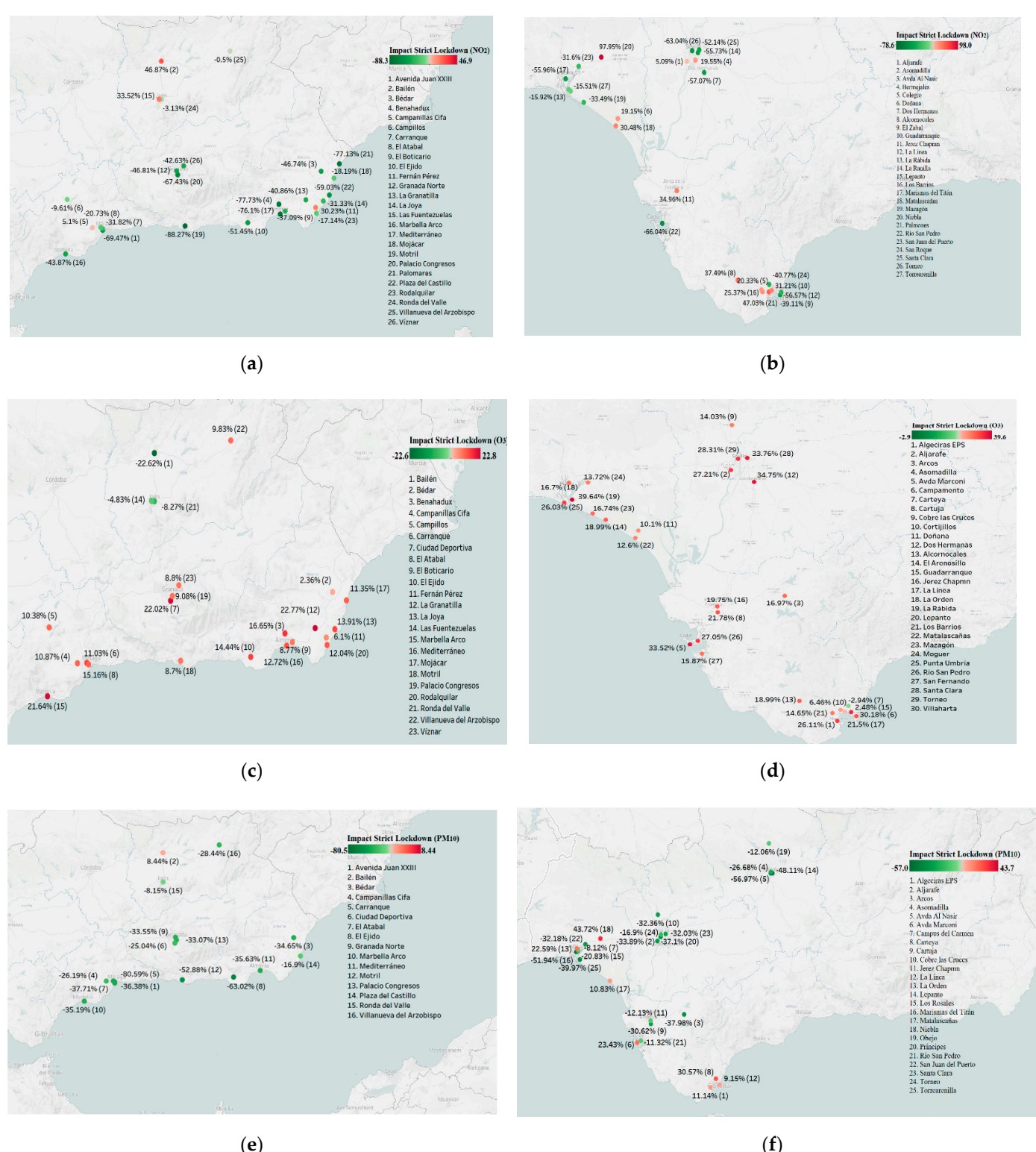

**Figure 3.** The left and right sides contain the Lockdown effect estimates by air monitoring stations in the SE and SW, respectively. (**a**) Strict lockdown (phase 0) impact on NO$_2$ in SE Andalusia; (**b**) Strict lockdown (phase 0) impact on NO$_2$ in SW Andalusia; (**c**) Strict lockdown (phase 0) impact on O$_3$ in SE Andalusia; (**d**) Strict lockdown (phase 0) impact on O$_3$ in SW Andalusia; (**e**) Strict lockdown (phase 0) impact on PM$_{10}$ in SE Andalusia; (**f**) Strict lockdown (phase 0) impact on PM$_{10}$ in SW Andalusia.

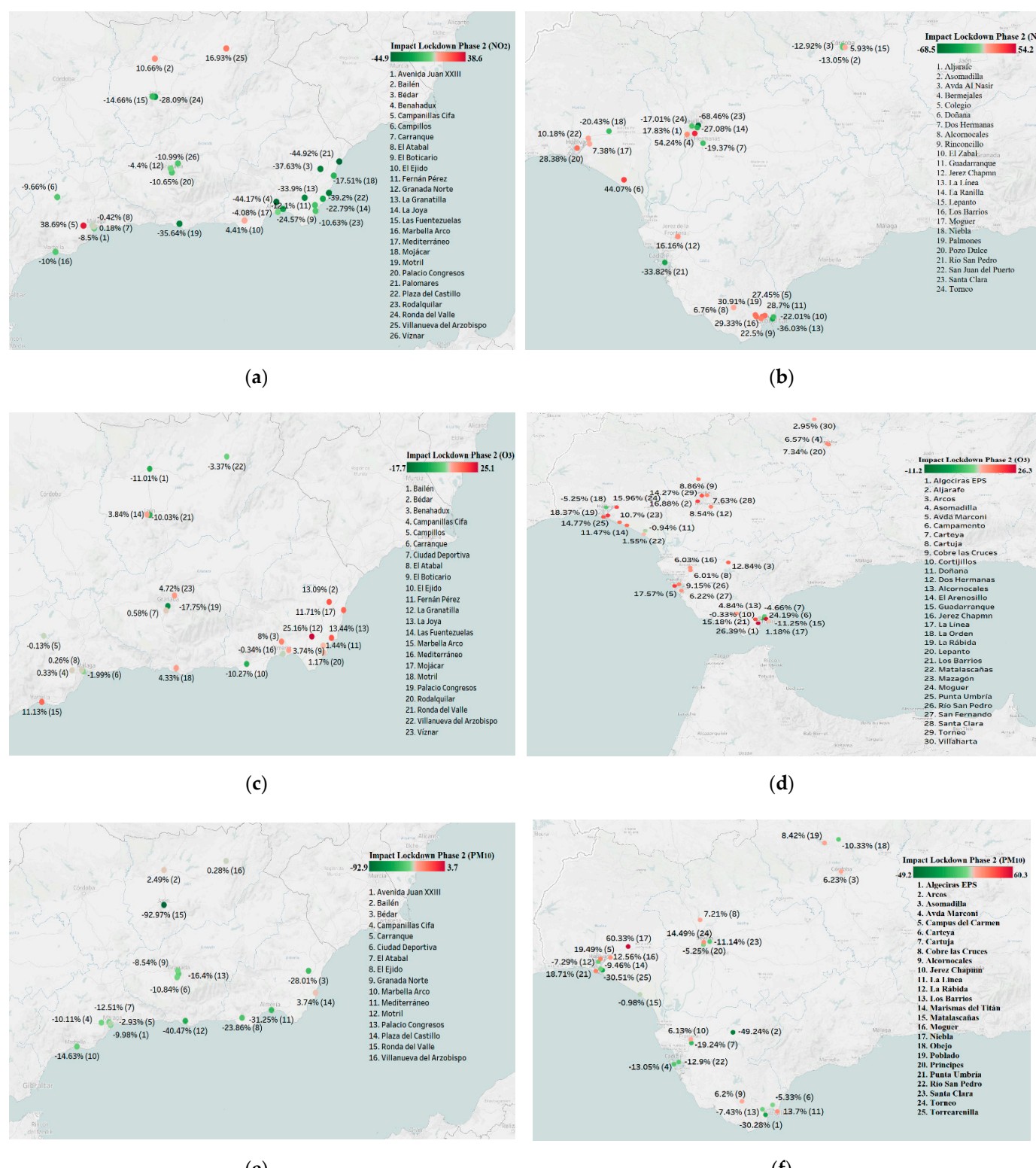

**Figure 4.** The left and right sides contain the estimates of the Lockdown (Phase 2) effect by air monitoring stations in the SE and SW, respectively. (**a**) Lockdown (phase 2) impact on NO$_2$ in SE Andalusia; (**b**) Lockdown (phase 2) impact on NO$_2$ in SW Andalusia; (**c**) Lockdown (phase 2) impact on O$_3$ in SE Andalusia; (**d**) Lockdown (phase 2) impact on O$_3$ in SW Andalusia; (**e**) Lockdown (phase 2) impact on PM$_{10}$ in SE Andalusia; (**f**) Lockdown (phase 2) impact on PM$_{10}$ in SW Andalusia.

The pattern in $O_3$ was opposed to $NO_2$ behavior during phase 0, presenting increments of tropospheric $O_3$ concentrations in the SW region (see Figure 3c), with a maximum registered value of 39.64% in La Rábida, Cordoba, and a slightly inferior increase in the SE, with a peak value of 22.02% in Ciudad Deportiva, Granada (see Figure 3d). During phase 2, the concentrations of $O_3$ increased in the SW, ranging from 1.18% in La Línea to 26.39% in Algeciras, while in the SE, they presented a remarkable decrement (−97% in Ciudad Deportiva; −49% in Marbella Arco) (see Figure 4c,d).

The behavior of $PM_{10}$ during phase 0 was heterogeneous, with a general decrease in the SE within a range from −8.15% (Ronda del Valle) to −80.59% (Carranque) and both positive and negative effects in the SW, with a peak reduction of 56.97% in Campus del Carmen (see Figure 3e,f). The concentration of $PM_{10}$ rose during phase 2 in both regions, with a weaker effect on the SE, where the number of particles decreased in all but three stations, Bailén among them, where ceramic factories resumed their activities, and Villanueva del Arzobispo, a station likely affected by the air pollution emitted by Bailén due to its proximity. The greatest drop was at Ronda del Valle station (92.97%), with a mean value for the other stations of less than −40% (−40.47% at Motril). In the SW, the decrease in $PM_{10}$ was more moderate, or the levels even increased, the highest positive effect occurring at Niebla station, in Huelva province (60.33%).

## 4. Discussion

This article estimates the COVID-19 induced lockdown's footprints on a battery of pollutants, including $NO_2$, $PM_{10}$, and $O_3$, in the Spanish community of Andalusia, one of the most affected by poor air quality due to its geographical location and climate conditions. The most affected pollutant by the complete lockdown period in March–June 2020 was $NO_2$, a pollutant emitted mainly through internal combustion engines of vehicles in urban areas. This fact suggests that improvements in air quality were triggered by the drastic reduction in road, sea, and air mobility within the reference period.

Consequently, SW Andalusia recorded a 32.7% drop in $NO_2$, which was even more prominent in the SE part, with a 46.7% plunge. These rates were below those shown in Madrid and Barcelona by Baldasano [26], who accounted for a 50−60% decrease in both cities compared to the pre-lockdown period. Moreda Piñeiro et al. [28] also estimated more intense falls in NOx levels at several air monitoring stations in A Coruña and Vigo. Cárcel-Carrasco et al. [29] also observed more pronounced decreases for Valencia, Bilbao, and Sevilla cities after March 14. Similarly, Querol et al. [31] documented very sharp drops, above 50%, in Valencia, Sevilla, Malaga, Badajoz, Madrid, and Valladolid cities, especially during the strictest phase of the lockdown, but even close to 35–43% in Sevilla, Barcelona, Madrid, Valencia, and Valladolid during the relaxation phase. Barcelona city also registered a reduction of 51% in $NO_2$ during the first month of the lockdown, according to Tobias et al. [25], and decreases were over 60% at various air monitoring stations of Valencia city [27].

Moreover, the pause in economic activities was more pronounced in metropolitan areas compared to rural regions, so falls in $NO_2$ pollution were more intense in the first ones. They reached their maximum values during the strict lockdown (phase 0) in some province capitals, such as Sevilla (−63.04%), Cadiz, (−66.04%), Cordoba (−78.62%), Malaga (−69.47%), Almeria (−77.73%), and Granada (−67.43%). These findings are in line with similar research by Briz-Redón et al. [30]. They calculated that $NO_2$ emission reductions were greater in highly populated cities such as Madrid, Barcelona, Sevilla, and Valencia. In contrast, the smaller city of Santander did not showcase significant changes through the same strict lockdown period.

The mean variation in $PM_{10}$ was less pronounced than $NO_2$, only −10.2% within Andalusia's territory. This pattern can be explained by the fact that particulate matter consists of hundreds of different chemical substances produced by a wide range of sources (road traffic, construction sites, smokestacks, wildfires, dry soil, and rock erosion that favor resuspension), but also by complex chemical reactions between $SO_2$ and NOx. Even if the

complete lockdown had a clear impact on direct sources of $PM_{10}$, its influence on indirect ones is more blurred. Moreover, biomass combustion episodes were minimal between March and June 2020 due to above average rainfall, mild temperatures—only days March 12 and 15 were chilly in SE Andalusia—and only a limited number of forest fires, mainly in Almeria province [79]. This result is in line with research on lockdown in the Community of Madrid [32], where decreases in $PM_{10}$ and $PM_{2.5}$ pollutants were less pronounced than those of NO and $NO_2$, as particulate matter is emitted in large quantities and tends to remain in the air for long periods.

Industrial activities are concentrated in specific SW areas in Andalusia and are highly polluting. These hotspots are Ria of Huelva and the Strait of Gibraltar, and to a lesser extent, some sites at Jaen (Bailén ceramic factories), Sevilla, Granada, and Almeria (Carboneras). The Huelva metropolitan area features an oil refinery, copper smelter, and combined-cycle power plant, and is close to the Riotinto mining district, based on massive pyrite deposits [80,81], while in the Strait of Gibraltar lies several major oil refineries, gas liquefaction plants, and petrochemical and metallurgical facilities [82]. Seaports are also present in both districts, and cargo handled by dockers causes the resuspension of loose materials, contributing to PM loads [63]. Consequently, $NO_2$ and $PM_{10}$ emissions fell during the total industrial production stoppage in the community's southwest: in the strictest lockdown (phase 0), $NO_2$ decreases were up to 19.5%, and $PM_{10}$ falls exceeded 21.5% when compared to the pre-lockdown period. Industrial activity was allowed to return when phase 2 was enacted, and $NO_2$ concentrations increased in Huelva, up to 28.38% at Pozo Dulce, and reached 30.91% at Palomares air monitoring station, located within the Strait of Gibraltar. $PM_{10}$ levels during the same lockdown phase were more moderate; they were up 60.33% at Niebla station (close to Ria of Huelva) and 13.7% at La Línea station (within the Strait of Gibraltar).

On the other hand, mobility restrictions during lockdown resulted in a noticeable increase in near-surface $O_3$ levels. Tropospheric ozone rose 9.3% in SW Andalusia and 20.6% in the SE half during the strict lockdown (phase 0). Its concentration went down 12%, and 7.2% in the respective areas as restrictions were gradually lifted in phase 2. This behavior is consistent with $O_3$ formation chemistry, produced by the photooxidation of volatile organic compounds in the presence of NOx and sunlight in a VOC-limited regime, common in urban areas. Lower emissions of NOx lead to inferior $O_3$ titration, and $O_3$ levels rise: $O_3$ increases were due to $NO_2$ reductions between March and June, when mobility restrictions were present. Additionally, $O_3$ concentrations rose the most in urban areas, resulting in a 39.64% hike in Sevilla, 33.76% in Sevilla, 33.52% in Cadiz, 37.36% in Cordoba, 22.02% in Granada, and 21.64% in Marbella cities. These results are in line with Betancourt-Odio et al. [32] on the Community of Madrid, where mean increases of 20.3% were recorded for $O_3$ during the lockdown. Hidalgo-García and Arco-Díaz [30] also estimated increases in $O_3$ pollution during the day and at night in Andalusia's coastal and inland urban nuclei. Tobias et al. [21] argues that the same relative to Barcelona city, and Mortell-Marugán [33] showed high increases in ozone levels (above 50%) in Spanish urban and suburban environments. Moreda-Piñeiro et al. [28] found that this pollutant rose between 5% to 16% in two Galician cities, and Briz-Redón et al. [30] estimated a positive trend in major Spanish cities too. However, it was only significant for Barcelona and Valencia. On the other hand, Donzelli et al. [27] argue that overall $O_3$ levels decreased in Valencia city during lockdown mainly due to weather conditions. Similarly, Ordoñez et al. [17] state that $O_3$ levels augmented in most parts of Europe, enhanced by relatively high temperatures, low humidity, and stronger solar radiation, while in the Iberian Peninsula they experienced decreases precisely because of higher specific humidity, lower solar radiation, and reduced wind speed.

Regular rises in the sunshine and mean daily temperatures between March and June triggered an increase in $O_3$ levels in Andalusia, as UV rays act as catalysts for reactions between NOx and CO emissions. Average temperatures in the community were above those during 2019 in almost every province, but Cordoba (19.6 °C) and Malaga (19.48 °C) particularly stand out. Precisely, the air monitoring stations of Marbella Arco (21.34%) in

Marbella and Lepanto (37.36%) in Cordoba registered the greatest of these $O_3$ increases during the strict lockdown (phase 0). On the other hand, the lowest mean temperature between June and March 2020 was recorded in Granada (19.98 °C), which was 1.6 °C below the mean temperature of Jaen, the second coolest site during those months. Low temperatures may cause pollution peaks related to thermal inversion, influencing air circulation in the lower atmospheric layers and preventing the dispersion of pollutants. Moreover, Granada's location in a valley surrounded by high mountains intensifies anticyclonic conditions and traps polluted air within the metropolitan area [71,72]. Therefore, $PM_{10}$ levels experienced mild decreases during the strict lockdown (phase 0) at Granada's air monitoring stations, not exceeding 34%. At Jaen's stations, $PM_{10}$ dropped by merely 28.44%, which contrasts with Malaga's 80.59% and Almeria's 63.02% plunges.

Several intense wind episodes were observed in southern Spain during the lockdown period, particularly at Almeria airport air monitoring station, which recorded maximum speeds in April (95 kph) and June (93 kph). Higher wind speed plays a significant role in the transportation and dilution of pollutants and is likely to improve wind quality. $NO_2$ reductions were more pronounced in Almeria during phase 2 compared to the rest of SE Andalusia, particularly at Palomares ($-44.92\%$) and Benahadux ($-44.17\%$) stations. Changes relative to this pollutant were more modest at Granada, Jaen, and Malaga, with maximum decreases of 35.64%, 28.09%, and 10%, respectively.

Another region affected by the strong winds during the reference period was Cadiz, reaching a speed of 139 kph. This event led to increased $O_3$ levels at rural stations as the wind carries $O_3$ from those distant regions responsible for $O_3$ generation, mainly traffic and industrial zones, and urban agglomerations. Thus, the highest variations in $O_3$ rates in SW Andalusia during phase 2 corresponded to stations of Cadiz province, especially Algeciras EPS (26.39%) and Campamento (24.19%), both located within the Strait of Gibraltar. Sevilla, Cordoba, and Huelva $O_3$-level increases did not exceed 16.88%, 7.34%, and 18.37%, respectively.

Additionally, wind enables Saharan dust intrusions, leading to elevated particulate matter levels. Andalusia is particularly exposed to these dust loads due to its proximity to the African continent and dry weather [51–53]. In 2020 Saharan dust episodes took place during March, April, May, and June, and affected both the SE and SW parts of the community. Specifically, African dust was recorded for 41 days in the SE region and 37 days in the SW.

Cadiz and Malaga provinces recorded the highest mean precipitation levels during our reference period, reaching 64.20 and 52.30 mm, respectively. Rainfall reduces the presence of pollutants in the atmosphere through soil deposition, which explains a more prominent decrease in $PM_{10}$ levels during phase 2 in Cadiz province, especially at Arcos ($-9.24\%$) and Algeciras EPS ($-30.28\%$) stations. This province's average precipitation was 0.9 mm in June, while the rest of Andalusia experienced no rainfall.

As far as rural areas are concerned, they were subject to $O_3$ increases during the lockdown's phases. During the strict lockdown, the rural stations of Benahadux and Mojácar recorded the highest rises in near-surface ozone (16.7% and 11.35%, respectively) in SE Andalusia. Both are located near the Carboneras industrial district and are likely influenced by winds carrying polluted air masses not only from this regional industrial zone, but also from the WMB's highly populated settlements and from the Atlantic Ocean via the Gibraltar Strait [53]. On the other hand, $O_3$ increases were more pronounced in the SW part, especially at Arcos (16.97%) and Cobre las Cruces (14.03%). Both lie within the Guadalquivir Valley, the first one between the bays of Algeciras and Cadiz, and the second one north of Sevilla city. The Guadalquivir Valley, dominated by southwestern winds that carry $O_3$ from the lower basin, is a natural channel for the transport and dispersion of polluted air from the Huelva towards the northeastern inner regions of the basin, where they mix with local traffic emissions in Sevilla's metropolitan area [54–58]. Its terrain is relatively homogenous and flat, which favor the penetration of polluted air loads.

PM$_{10}$ levels rose only at Matalascañas (10.38%) and Alcornocales (6.2%) rural stations during the complete lockdown period, which may be attributed to the migration of polluting particles from the Huelva industrial district, the Gibraltar Strait, and the Cadiz harbor, respectively, as the model controls for Saharan dust intrusion and biomass combustion [52,64,80,81].

## 5. Conclusions

The present article assessed the footprint of COVID-19 lockdowns on a set of pollutants in southern Spain using a multivariate linear regression model. Each analyzed pollutant showed a different pattern during the lockdown period: average NO$_2$ decreases were 39.7%, PM$_{10}$ fell only 9.9%, and O$_3$ levels increased by 3.7%. Our dataset covered the period from 1 January 2017 to 31 December 2020, including the entire length of the Spanish lockdown, comprising the period between 15 March 2020 and 20 June 2020, decreed by the Spanish government to flatten the epidemic's infection curve. One of the significant contributions presented in this article is the heterogeneous footprint of the lockdown period on pollution patterns across geographic regions. By employing data from 80 air monitoring stations, 27 in SE Andalusia and 53 in the SW half, and the three different pollutants, the article aims at monitoring such diversity.

According to our results, NO$_2$ levels were reduced more prominently in major urban areas where heavy road traffic was present before the pandemic. Mobility restrictions managed to lower this pollutant to 78.62% in Cordoba and 77.73% in Almeria during the strict lockdown (phase 0). Moreover, the lockdown effect was more intense in the community's SW part, as most industrial districts and heavy traffic seaports lie in Huelva and the Strait of Gibraltar. These sites were precisely those that swiftly increased their NO$_2$ emissions during phase 2, when economic activities were reactivated. Pollution rose to 28.38% at Pozo Dulce (Huelva) and 30.91% at Palomares (Strait of Gibraltar, Cadiz province).

Heterogeneity between urban and peripheral areas was more moderate relative to PM$_{10}$ owing to other factors, such as topography, the intrusion of Saharan dust, biomass burning, or climate conditions such as temperature, wind, and rainfall. In Granada, only a mild reduction was recorded in PM$_{10}$ levels during the strict lockdown (phase 0). The province featured the lowest mean temperatures, and pollutants remained trapped in the area because of its bowl-like topography. On the contrary, regions characterized by higher precipitation and wind speed, as at Cadiz, enjoyed a larger and more significant improvement in particulate matter pollution than the rest of the community, even in the presence of Saharan dust loads.

On the other hand, the lockdown exerted a positive effect on O$_3$ concentration in Andalusia, which rose by 14.95% during the strict lockdown (phase 0). Lower O$_3$ titration caused by reductions in NOx and VOCs during mobility restrictions was exacerbated by Andalusia's higher temperatures and insolation, compared to the rest of Spain. Cordoba and Malaga were noticeably more affected by more intense sunlight between March and June 2020, and hence O$_3$ increases were more prominent.

Our results offer several contributions for environmental policymakers focused on enhancing air quality within Andalusia's territory. First, the autonomous community's location in the WMB, the concentrations of significant industries in particular areas of the region, and climate heterogeneity should be considered when drafting effective air pollution reduction plans. Therefore, policies ought to distinguish between generic pollution in the Andalusian territory and those critical sites where additional measures need to be implemented to achieve decent air quality. Among these critical hotspots, we include the industrial districts of Gibraltar (Cadiz), Sevilla and Granada's metropolitan areas, and the coast of Malaga and Almeria. Low emission zones (LEZ) could be introduced in regions that are heavily affected by road traffic, such as the metropolitan areas of Sevilla, Granada, and Malaga. According to Law 7/2021 on Climate Change and Energy Transition, it will be mandatory for Spanish municipalities above 50,000 inhabitants to introduce at

least one LEZ in their urban areas by 2023 [83,84]. Out of the 149 Spanish municipalities that need to implement these measures to reduce road traffic emissions, 29 are located in Andalusia (Alcalá de Guadaira, Algeciras, Almeria, Benalmádena, Cadiz, Chiclana de la Frontera, Cordoba, Dos Hermanas, El Ejido, El Puerto de Santa María, Estepona, Fuengirola, Granada, Huelva, Jaen, Jerez de la Frontera, La Línea de la Concepción, Linares, Malaga, Marbella, Mijas, Motril, Roquetas de Mar, San Fernando, Sanlúcar de Barrameda, Sevilla, Torremolinos, Utrera, and Vélez Málaga). Nonetheless, these projects have been delayed by mayors due to their low popularity among diesel car owners, and so far only two cities have effectively introduced LEZ, Madrid and Barcelona. The city councils of major Andalusian agglomerations such as those of Sevilla, Malaga, Granada, and Cordoba are already working on their LEZ projects that will be progressively implemented through 2023 and will receive funding from Next Generation EU. On the other hand, energy audits could be performed in the large industrial agglomerations at Ria of Huelva, Strait of Gibraltar (Cadiz), and Carboneras (Almeria) to monitor NOx emissions caused by hydrocarbon combustion and refining. Furthermore, biomass burning (of stubbles and boilers) days could be restricted at rural sites to control mass steams filled with $NO_2$ and $PM_{10}$.

A limitation of this research is associated with the location dispersion of the automatic monitoring stations and the poor performance of the manual air quality monitoring stations. A more significant number of monitoring stations would allow for better capturing behavior patterns between geographic areas. In addition, for the $PM_{2.5}$ pollutant, during the year 2021, all manual stations were below 55% data capture, so they could not be considered for the study, and we decided not to study the $PM_{2.5}$ pollutant.

**Author Contributions:** Conceptualization and methodology, M.A.B.-O. and M.C.-G.; software, validation, and formal analysis, M.A.B.-O., M.C.-G., A.Z.-G. and E.W.; investigation and writing—original draft preparation, M.A.B.-O., M.C.-G., A.Z.-G. and E.W.; writing—review and editing, M.A.B.-O., M.C.-G., A.Z.-G. and E.W. All authors have read and agreed to the published version of the manuscript.

**Funding:** This research received no external funding.

**Institutional Review Board Statement:** Not applicable.

**Informed Consent Statement:** Not applicable.

**Data Availability Statement:** The Atmospheric Quality Area—Air Network of the autonomous region and the Atmospheric Protection Service of Andalusia issued the data. In addition, meteorological data were downloaded from the Open Data platform of the State Meteorological Agency (AEMET) via its Application Programming Interface (API).

**Conflicts of Interest:** The authors declare no conflict of interest.

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
