# Peer review of "Footprints of COVID-19 on Pollution in Southern Spain"

_atmosphere, doi:10.3390/atmos13111928_

Round 1

Reviewer 1 Report

The manuscript, “Footprints of COVID-19 on pollution in southern Spain,” written by Eszter Wirth, Manuel Alejandro Betancourt-Odio, Macarena Cabeza-García, Ana Zapatero-González concern the impact of COVID-19 lockdowns on pollutants PM10, NO2 and O3 at urban, suburban, and rural stations in the Andalusia. The authors employed multiple regression techniques to model air pollution based on a variety of explanatory factors: meteorological variables, biomass combustion by wildfires, Saharan dust episodes, weekend effect, and, above all, the several phases of the nationwide lockdown period enforced by the Spanish government since March 15 to June 20, 2020. The authors claimed that the lockdown’s footprint was diverse across regions and each analyzed pollutant showed a different pattern during the lockdown period. The manuscript focuses on a topical and important issue.

The following changes are required before to publication:

- section Keywords: it should be mention about air pollution and kind of pollutants.

- Line 45 –“Thus, many research papers have been published on a global scale to monitor the impact of Covid-19 induced lockdowns on air quality, mainly focusing on large urban 44 settlements [6-22]” - It is not professional to make such a reference to a bibliography;

- Line 52- “Research Articles also focused on major Spanish cities,…” articles should be with a lowercase letter

- Line 54 – It is not professional to make such a reference to a bibliography,

-        - Line 209, fig.1. – should be: a) Andalusia zoning map, b) Distribution of air quality monitoring stations.

-        - Tables are usually signed at the top of them.

Author Response

We thank the referee for these valuable comments. We have taken it into account to improve the paper.

Reviewer 2 Report

The authors assessed the levels of NO2, O3, and PM10 in Andalusia within the 2017-2020 period using a multivariate linear regression model. They found a roughly decrease for NO2 and PM10 during the lockdown, two pollutants mainly from the primary sources. However, O3 showed increasing trend during the lockdown. The result is interesting and I have some minor comments.

Major concern

I found it is interesting that the author find the ozone concentration increased when NO2 concentration decreases in many regions like La Rábida, Cordoba, “Line 492-498”. The author explained it in Line 596-597, that mainly due to a decreased NO titration for O3. However, O3 was mainly formed from the photolysis of NOx in the prence of VOCs oxidation rather than “produced by the photooxidation of volatile organic compounds in the presence of NOx in line 596”. Could that due to a rising VOCs emission because we know O3 could be limited by VOCs or NOx, and it is possible more VOCs could be released since more cooking activies, or other would happen during the lockdown.

Minor concers:

Line 31, please also mention the year after “late February”;

Line 327, please mention the “Table ?”

In Table 1, please change the “,” to “.’ for the numbers.

Author Response

We thank the referee for these valuable comments. We have taken it into account to improve the paper. Following your advice, most parts of the article were rewritten, considering the comments made.

Reviewer 3 Report

General comment:

Authors made a good statistical analysis of pollution concentrations in southern Spain. The paper is in the scope of the journal and has scientific soundness. The graphics are well-prepared and the language style is also very good. Therefore in my opinion it should be published in the journal after some major corrections. I hope the paper will be shared with local authorities for the planning of a sustainable future with clear air. 

One of my concerns is that the research itself is like the background of the whole story.  I encourage authors to put more effort to make research part of the first plan considering maybe reducing some other parts. On the other hand, it would be good to add information about air quality standards in Spain. The second concern is related to the design of the research as authors are trying to find patterns in the whole Andalusia region using very sparse measurement points coverage. The air pollution indication, in this case, may be sometimes very affected by various ultra-local conditions. To be clear I'm not saying that the research is designed wrongly, but the information about this limitation should be included. The third one is related to the metofactors and localization. Authors are saying that specific region localization etc, but what exactly? From my knowledge in Andalusia, there are mountains, valleys, and coastline, so I would expect that there will be noticeable differences between those regions, but it's not very clearly discussed in the text. The research discussing this issue in covid context can be found in the Scientific Reports paper about The influence of meteorological factors and terrain on air pollution concentration and migration by Weglinska et al. Beside different climates can you see any similarities in the context of terrain etc? 

Another comment is related to external dust sources like combustion and dust from Africa. I would expect something more quantitative, did you try to find for example radiometric analysis of pollution in your region or at least a calculation of  P2.5/PM10 ratio like Xu, G. et al. Spatial and temporal variability of the PM2.5/PM10 ratio in Wuhan. Cent. China. Aerosol Air Qual. Res. 17?

The last one is related to the conclusions. Are you sure that there was enough measurement points coverage in the analyzed cities to conclude about low emission zones like in line 722?

 Specific comments:

1. Abstract is very well written.

2. Introduction 

Please do not use combined citations like 6-22. If all papers are important please mention why for each or for groups of 2-3 not 16 in one sentence.

Line 50 - please add a reference.

What means more modest ameliorations? Not only the development is important here. There are more factors here. For example study in Europe showed that household heating with coal is the main source of air pollution in Poland, so covid had not has much to reduce besides the transportation impact see the paper from Krakow in the special issue of  Air Quality in a Changed World: Regional, Ambient, and Indoor Air Concentrations from the COVID to Post-COVID Era (III) of Aerosol and Air Quality Research Journal (https://aaqr.org/articles/special-issues/post-covid  and clarify this paragraph. 

2. Methodology and results

What is the percentage of individual sources for PMs and gases?

Are these (pollution) measurements reference ones or not? What are the uncertainties? Please clearly specify it in the text. 

How were the meteorological factors of different scales and units considered in the regression? Is rain an important factor in this region? It's surprising that in 379 linie rain has almost no impact, especially on PM10 level. 

How did the authors include biomass combustion and dust in variable X? Is there any data provider of combustion episodes or sth like this?  

Author Response

(The authors gave the same response as above.)

Round 2

Reviewer 3 Report

The authors addressed all my comments and made major corrections, improving the paper significantly. Now I can recommend the article to be accepted for publication.